# Exercise Training Upregulates Cardiac *mtp* Expression in *Drosophila melanogaster* with HFD to Improve Cardiac Dysfunction and Abnormal Lipid Metabolism

**DOI:** 10.3390/biology11121745

**Published:** 2022-11-30

**Authors:** Tianhang Peng, Meng Ding, Hanhui Yan, Qiufang Li, Ping Zhang, Rui Tian, Lan Zheng

**Affiliations:** Key Laboratory of Physical Fitness and Exercise Rehabilitation of Hunan Province, Hunan Normal University, Changsha 410012, China

**Keywords:** *Drosophila*, *mtp*, cardiac dysfunction, exercise training, lipid metabolism, β-oxidation

## Abstract

**Simple Summary:**

It is well-established that the heart regulates systemic lipid metabolism, but the exact molecular mechanisms remain largely unclear. A high-fat diet (HFD) can lead to systemic lipid overload and a range of cardiac dysfunctions such as arrhythmias, fibrillation and reduced contractility. Although pharmacological treatment is often effective, there are always side effects. Accordingly, exercise training represents an effective alternative intervention to medication. In this experiment, we established a high-fat *Drosophila* model by HFD feeding and conducted exercise training intervention to investigate whether the exercise intervention altered the expression of the target gene (*mtp*), thus affecting systemic lipid metabolism and cardiac function. Our results suggest that specific knockdown of *mtp* mitigates HFD-induced elevation of systemic triglycerides and protects cardiac contractility to some extent. Further analysis showed that exercise training could upregulate the expression of *mtp* to restore dysregulated systemic lipid metabolism and cardiac function induced by HFD. Overall, we explored the important role of *mtp* in systemic lipid metabolism and cardiac function, providing new insights for future clinical studies related to lipotoxic cardiomyopathy and the potential use of exercise training to treat abnormal lipid metabolism and cardiac dysfunction due to HFD.

**Abstract:**

Current evidence suggests that the heart plays an important role in regulating systemic lipid homeostasis, and high-fat diet (HFD)-induced obesity is a major cause of cardiovascular disease, although little is known about the specific mechanisms involved. Exercise training can reportedly improve abnormal lipid metabolism and cardiac dysfunction induced by high-fat diets; however, the molecular mechanisms are not yet understood. In the present study, to explore the relationship between exercise training and cardiac *mtp* in HFD flies and potential mechanisms by which exercise training affects HFD flies, *Drosophila* was selected as a model organism, and the GAL4/UAS system was used to specifically knock down the target gene. Experiments revealed that HFD-fed *Drosophila* exhibited changes in body weight, increased triglycerides (TG) and dysregulated cardiac contractility, consistent with observations in mammals. Interestingly, inhibition of cardiac *mtp* expression reduced HFD-induced cardiac damage and mitigated the increase in triglycerides. Further studies showed that in HFD +*w^1118^*, HFD + *Hand > w^1118^*, and HFD+ *Hand > mtp^RNAi^*, cardiac *mtp* expression downregulation induced by HFD was treated by exercise training and mitochondrial β-oxidation capacity in cardiomyocytes was reversed. Overall, knocking down *mtp* in the heart prevented an increase in systemic TG levels and protected cardiac contractility from damage caused by HFD, similar to the findings observed after exercise training. Moreover, exercise training upregulated the decrease in cardiac *mtp* expression induced by HFD. Increased Had1 and Acox3 expression were observed, consistent with changes in cardiac *mtp* expression.

## 1. Introduction

Unhealthy eating habits cause an excessive accumulation of lipids in the body (i.e., “obesity”), which represents a high-risk factor for harmful effects on metabolism and heart function and can lead to the development of a range of diseases such as heart disease, type 2 diabetes and cancer. After hydrolysis into glycerol and fatty acids, fats are packaged in the endoplasmic reticulum of small intestinal cells as chylomicrons (CMs) [1,2] and the liver cells as very low-density lipoproteins (VLDLs) [3]. Microsomal triglyceride transfer protein (*mtp*) is involved in the assembly of lipoproteins, transferring lipids to lipoprotein B (apoB) [4], which allows apoB to fold correctly and assemble primitive lipoprotein particles involved in systemic lipid metabolism [5]. An increasing body of evidence suggests that mice lacking *mtp* in the liver or intestine exhibit systemic or specific reductions in plasma triglyceride and cholesterol levels [6,7]. Studies on humans have shown that *mtp* is associated with lipid droplets in brown and white fat and plays an important role in lipid droplet formation and/or turnover [8]. In addition, *mtp* is composed of α and β subunits, and mitochondrial β-oxidation is the main metabolic system for the catabolism of fatty acids, generating the main energy source for various cellular processes. β-oxidation catalyzes fatty acids through four reactions (dehydration, oxidation, hydration, and thiolysis) to produce acetyl coenzyme A. Importantly, *mtp* catalyzes the last three steps of β-oxidation of long-chain fatty acids. The heart can influence mitochondrial β-oxidation by regulating the expression of *mtp*, which affects systemic lipid metabolism.

Exercise training is recognized as one of the most effective treatments for obesity and cardiovascular disease. In mouse studies, exercise training improved HFD-induced glucose intolerance and insulin resistance while increasing acetylcholine levels, ChAT activity and PKC activity [9]. It could also inhibit inflammation in the adipose tissue of obese mice induced by a high-fat diet [9,10]. Several human metabolic diseases, including obesity, impaired glucose tolerance and type 2 diabetes, have been advocated for prevention and treatment with exercise training [11,12,13,14].

Unlike other animal models such as *Mice* and *Zebrafish, Drosophila* has a short life cycle (from egg to adult in about 10 days), produces many offspring (it can produce 30 generations a year, is convenient to feed and can be used for genome editing. More importantly, sequencing and annotation of the *Drosophila* genome have shown that genes involved in the normal development of several organs, including the heart, are highly conserved and less redundant than in vertebrates. Moreover, it has been shown that a large number of human disease genes have *Drosophila* counterparts and that their hearts exhibit developmental and functional similarities to the vertebrate heart [15,16,17]. In addition, several models of HFD-induced obesity have been successfully developed in *Drosophila*, encapsulating the distinctive features of human obesity and diabetes [18].

An increasing body of evidence from recently published studies has confirmed that the heart plays an important role in regulating systemic lipid metabolism [19]. Animal experiments showed that cardiac MED13 expression regulation could control systemic lipid metabolism [20]. However, the specific mechanisms warrant further exploration. By establishing a *Drosophila* model of HFD and conducting cardiac function measurements [21,22,23], we investigated how exercise training affects HFD-induced cardiac disorders and abnormalities in lipid metabolism. Our findings suggest that specific knockdown of *mtp* in the heart reduces the elevation of systemic TG and alleviates HFD-induced cardiac dysfunction. In addition, in HFD + *w^1118^*, HFD + *Hand > w^1118^*, and HFD + *Hand > mtp^RNAi^*, exercise training could restore decreased cardiac *mtp* expression induced by HFD and improve mitochondrial β-oxidation capacity in cardiomyocytes. Our findings improve current understanding of the role of *mtp* on systemic lipid metabolism and cardiac function and provide new insights for future clinical studies related to lipotoxic cardiomyopathy and the potential use of exercise training to treat abnormal lipid metabolism and cardiac dysfunction induced by HFD.

## 2. Materials and Methods

### 2.1. Drosophila Strains and Husbandry

*w^1118^ Drosophila* were obtained from the Bloomington *Drosophila* Stock Center, and *UAS-mtp^RNAi^ Drosophila* from the *Drosophila* Center, Tsinghua University. To reduce *mtp* expression in the heart, we used *Hand-Gal4* (cardiomyocyte-specific driver BL48396). The normal food diet (NFD) consisted of a combination of yeast, maize and starch; the high-fat food diet (HFD) was made from 30% coconut oil mixed with the 70% volume of NFD [24,25]. All fruit flies were fed with NFD for five days, then high-fat group fruit flies were fed HFD for five days.

*Hand-Gal4* male flies were crossed with *w^1118^* female flies and female flies of the F1 generation were collected within 12 h of eclosion. These flies and virgin flies of *w^1118^* underwent different interventions (NFD +*w^1118^*, HFD + *w^1118^*, HFD + E + *w^1118^,* NFD + *Hand > w^1118^*, HFD + *Hand > w^1118^*, and HFD + E + *Hand > w^1118^* (E is for exercise)). *Hand-Gal4* male flies were crossed with *UAS-mtp^RNAi^* female flies, and female flies from the F1 generation were eclosion was collected within 12 h of eclosion. These flies were defined as cardiomyocyte-specific knockdown of *mtp* groups and were subjected to different interventions (NFD + *Hand > mtp^RNAi^*, HFD + *Hand > mtp^RNAi^* and HFD + E + *Hand > mtp^RNAi^*). A total of 1800 virgin flies that were 10 days old were obtained (200 in each group). All flies were placed in transparent glass tubes (20–30 per tube). NFD-fed flies were placed in a constant temperature and humidity incubator (25 °C, 50% humidity, 12 h day-night cycle). HFD-fed flies were reared in an environment of 21–22 °C to prevent part of the high-fat food from melting and causing the flies to stick to the food and die.

### 2.2. Heart Function Tests

Based on previous studies, flies were anesthetized with FlyNap (Triangle Biotechnology, Shanghai, China), and the dissected fly hearts were preserved in oxygenated saline. The heartbeats were captured using an EM-CCD high-speed camera (130 fps, 30-s video), and heart rate, cardiac period, arrhythmia, contractility, shortening fraction, etc., were analyzed using semi-automatic optical heartbeat analysis software (SOHA) [26,27].

### 2.3. Body Weight and Triglyceride (TG) Measurement

An electronic microbalance (Shimadzu, AUW220D, Kyoto Japan) was used to weigh the fruit flies and record the weight of each fly for analysis.

TG concentration was measured using a triglyceride (TG) assay kit (Nanjing Jiancheng Bioengineering Institution, Nanjing, China) according to the manufacturer’s instructions [28].

### 2.4. Quantitative Real-Time Fluorescence PCR (qPCR)

Fifty hearts were placed into 1 mL of Trizol reagent lysis solution for homogenization and RNA extraction. Trizol was used to extract the organic solvent, and 10 μg total RNA was purified using oligo (dT) synthesized from total RNA with superscript II reverse transcriptase (Invitrogen). qPCR amplification reactions were performed in triplicates by mixing 1 μL of RT product with 10 μL of SYBR qPCR Mastermix (TaKaRa) containing the appropriate PCR primers. Thermal cycling and fluorescence monitoring were performed in an ABI7300 (Applied Biosystems, United States) using the following PCR conditions: (30 s at 95 °C, 5 s at 95 °C, and 30 s at 60 °C) × 40. Values were normalized with rp49. Primers used were as follows: rp49(the reference gene)F: 5′-CTAAGCTAGTCGCACAAATAGG-3′R; 5′-AACTTCTTAGAATCCGGTAGGG-3′*mtp*F; 5-ACGGAAATCCAGCAGAACACT-3′R:5′-ATACGTAAAGCCAACGGCCA-3′Acox3F; 5′-ACTTCCGTAGCGGACCTTTAG-3′R:5-GCAGAAGATAGTAGGGGTTCCA-3Had1F; 5-CTAAATAGCAAGGTACGCGGC-3′R; 5-GATAGTAGGGCCGTAGCGATAA-3′.

### 2.5. Oil Red O Dye Staining

Flies were anesthetized and dissected in ice-cold PBS. The ventral cuticle was removed from the abdomen using microscissors, and the viscera and genitalia were excised, leaving intact abdominal fat bodies attached to the dorsal cuticle. The plates were fixed with 4% paraformaldehyde for 20 min and washed three times with PBS for 10 min each time. The oil Red O solution (3:2 for staining solution and distilled water) was filtered, then incubated at room temperature for 30 min and washed three times with PBS. The photographs were taken under a Leica stereomicroscope, and after processing the images using Adobe Photoshop, the staining area of oil red was calculated using ImageJ software (ImageJ2,LOCI, Bethesda, MD, USA) [29].

### 2.6. Sports Training Devices and Programs

Consistent with previously established protocols, all fruit flies were fed under NFD conditions for 5 days, and those in the locomotor training group were trained from day 6 for 5 days, ending with training on day 10. *Drosophila* exercised for 1.5 h per day for 5 days [30,31,32]. During the locomotor training intervention, *Drosophila* spent 1.5 h in a glass tube without food. The fruit flies in the locomotor training control group were also placed in the glass tubes without food for 1.5 h, thus ensuring that all flies were in the same environment to avoid affecting their feeding rate [33].

### 2.7. Statistical Analysis

Figures were generated using GraphPad Prism6 software. Analyses were performed using the Statistical Package for the Social Sciences for Windows (SPSS) version 21.0 (SPSSInc., Chicago, IL, USA) [30,31,32]. All data are expressed as SEM ± mean. One-way ANOVA was used to identify differences between NFD, HFD and HFD + E groups of *Drosophila* with the same genetic background. Independent sample t-tests were used to identify differences between the two groups. A two-tailed *p*-value < 0.05 was statistically significant.

## 3. Results

### 3.1. HFD Causes Abnormal Lipid Metabolism and Cardiac Dysfunction in Drosophila

First, we established a *Drosophila* high-fat model. Female *Drosophila* were fed on NFD for 5 days and then on HFD for 5 days [24], and morphological photographs of *Drosophila* were taken (Figure 1A). Compared to the NFD group, the HFD-fed *Drosophila* exhibited a larger body size and a bulging abdomen with significantly higher body weight than the NFD group (Figure 2A). Moreover, the wing and ovary sizes of NFD and HFD flies were comparable. These findings suggested lipid uptake was responsible for the apparent abdominal bulge in HFD flies (Figure 1B–D). In addition, the oil red O staining area (Figure 2B,C) and TG content (Figure 2D) of abdominal fat were significantly greater in the HFD group of *Drosophila* than in the NFD group, demonstrating that our high fat protocol successfully induced a model of diet-induced obesity in *Drosophila*.

To investigate whether cardiac *mtp* is involved in obesity-induced cardiac dysfunction, we measured various cardiac function parameters and *mtp* mRNA expression in *Drosophila* NFD and *Drosophila* HFD, including heart rate (HR), cardiac cycle or heart period (HP), arrhythmia (AI), diastolic interval (DI) and systolic interval (SI), fibrillation (FL), diastolic diameter (DD), systolic diameter (SD), and fractional shortening (FS). We found varying degrees of increase in HR, AI and FL (Figure 3A,C,F) under HFD compared to NFD. It is well-established that fibrillation is the most common sustained arrhythmia and a high-risk factor for morbidity and mortality due to obesity [34]. Significant decreases in HP,SI, DD and FS (Figure 3B,E,G,I) and diastolic diameter and fractional shortening indicated a decrease in myocardial contractility in *Drosophila*. No significant differences were observed in DI and SD (Figure 3D,H). This finding suggests that HFD can lead to an increased heart rate in *Drosophila* and a concomitant increase in the frequency of arrhythmias and fibrillations, shorter cardiac periods and decreased pumping capacity of the heart (shortened fractions and diastolic diameters). Indeed, increased levels of fat and circulating blood lipids lead to cardiac dysfunction. For example, increased expression of lipid transport proteins in the heart leads to elevated lipids in cardiac myocytes with cardiac dysfunction [35]; cardiac pathological remodeling (i.e., hypertrophy and fibrosis) occurs during chronic HFD feeding, which affects systolic and diastolic function in the hearts of obese mice [36], consistent with findings of previous studies of lipotoxic cardiomyopathy [37]. Moreover, we found a significant decrease in *mtp* expression in the hearts of *Drosophila* in the HFD group compared to the NFD group (Figure 4A). These results suggest that increased dietary fat leads to obesity and cardiac dysfunction in *Drosophila*, which may be related to reduced *mtp* in cardiac myocytes (Figure 4A,B).

### 3.2. Specific Knockdown of mtp in the Heart Alleviates HFD-Induced Cardiac Dysfunction, Similar to the Effect of Exercise Training

We subjected high-fat *Drosophila* to five days of exercise and then measured *mtp* mRNA expression levels, cardiac function and whole-body triglyceride content. The results showed that exercise training significantly increased cardiac *mtp* expression in all groups of HFD *Drosophila* (Figure 5A). After exercise training, compared to HFD + *w^1118^* and HFD + *Hand > w^1118^*, HFD + E+*w^1118^* and HFD + E+*Hand > w^1118^ Drosophila* showed slower heart rate, increased cardiac period, decreased arrhythmia index, increased diastolic diameter, decreased systolic diameter, reduced fibrillation and increased shortening fraction (Figure 6A–C,F–I), with significant improvements in both cardiac rhythm and systolic function, effectively treating HFD-induced cardiac dysfunction. Exercise training also significantly reduced whole-body TG levels in HFD + *w^1118^* and HFD + Hand > *w^1118^ Drosophila*, demonstrating that exercise training could upregulate *mtp* expression in cardiac myocytes while modulating whole-body lipid levels and reducing whole-body TG, thereby protecting the heart from HFD-induced dysfunction.

HFD deteriorated cardiac function in *Drosophila* and caused a significant decrease in mRNA levels of *mtp* in the heart. Next, to investigate the relationship between HFD-induced cardiomyopathy and cardiomyocyte *mtp*, we targeted KD against *mtp* in cardiomyocytes and assessed whole-body TG levels in *Drosophila* with HFD. We found no significant difference in mRNA levels of *mtp* between NFD + *Hand > mtp^RNAi^* and HFD + *Hand > mtp^RNAi^* (Figure 5A). Moreover, the knockdown of *mtp* in the heart prevented an increase in systemic TG levels (Figure 6J), suggesting that *mtp* in cardiomyocytes plays an important role in regulating systemic lipid metabolism. Interestingly, no significant differences were found between the NFD + *Hand > mtp^RNAi^* group and the HFD + *Hand > mtp^RNAi^* group in DD, SD, FS and FL (Figure 6F–I); and differences were found in HR, HP, AI, DI and SI (Figure 6A–E) during M-mode ECG. These results suggest that NFD + *Hand > mtp^RNAi^* and HFD + *Hand > mtp^RNAi^* differed significantly, indicating that HFD mainly affected the cardiac rhythmicity of *Hand > mtp^RNAi^*. Compared to the control HFD + *w^1118^* and HFD + *Hand > w^1118^* groups, knocking down the cardiac *mtp* of HFD + *Hand > mtp^RNAi^* could reduce the damage of myocardial contractile function by HFD to some extent (Figure 5B). Knockdown of the cardiac *mtp* alone yielded effects similar to exercise training in treating cardiomyopathy in *Drosophila* with HFD, whereas the HFD + *Hand > mtp^RNAi^* group of *Drosophila* showed a significant increase in cardiac *mtp* expression (Figure 5A) and systemic TG levels (Figure 6J) after exercise training. We also found a decrease in HR (Figure 6A) and AI (Figure 6C) and an increase in HP, DI and SI (Figure 6B,D,E), indicating that the abnormal heart rhythm caused by HFD was significantly improved and the cardiac function of *Drosophila* was restored.

### 3.3. Exercise Training Upregulates Cardiomyocyte mtp Expression and Reverses the HFD-Induced Decrease in Cardiac Beta-Oxidation Capacity

It is well-established that β-oxidation catalyzes the degradation of long-chain fatty acids through four steps (dehydration, oxidation, hydration, and thiolysis) to produce acetyl coenzyme A. *mtp* consists of a hetero-octamer of four *mtp*α and four *mtp*β subunits. The *mtp*α subunit has long-chain 3-enoyl coenzyme A hydratase and long-chain 3-hydroxy coenzyme A dehydrogenase activities, catalyzing the second and third steps, respectively; the *mtp*β subunit has long-chain 3-ketoacyl coenzyme A Thiolase activity, catalyzing the fourth step. This study was based on previous approaches to β-oxidation function as assessed by acyl-coenzyme A dehydrogenase (β Hydroxy acid dehydrogenase 1 abbreviated as Had1) and 3-hydroxyacyl coenzyme A dehydrogenase (Acyl-CoA oxidase 3 abbreviated as Acox3) [38,39,40]. As seen in Figure 7A, HFD caused a decrease in Had1 expression in *w^1118^* and *Hand > w^1118^ Drosophila* cardiomyocytes, as well as a decrease in Acox3 expression in *w^1118^*, *Hand > w^1118^* and *Hand > mtp^RNAi^ Drosophila* cardiomyocytes, indicating that HFD can cause a decrease in cardiac β-oxidation and a reduction in the ability to catalyze long-chain fatty acids, predisposing to excessive accumulation of fat in the heart. However, after exercise training, there was a general increase in Had1 and Acox3 in *w^1118^*, *Hand > w^1118,^* and *Hand > mtp^RNAi^ Drosophila* cardiomyocytes, suggesting that exercise training could reverse the decrease in cardiac β-oxidation induced by HFD (Figure 7B), consistent with changes in cardiac *mtp* expression (Figure 5A).

## 4. Discussion

Unhealthy lifestyle habits, especially excessive intake of high-fat foods, represent a risk factor that predisposes to cardiovascular disease. *Drosophila* is widely acknowledged as an excellent model for studying cardiovascular function. More importantly, 77% of human disease genes have *Drosophila* counterparts, 26 of which have been associated with cardiovascular disease [15,16,17]. Exercise training is an effective means of preventing and treating obesity and associated cardiovascular disease [11,12,13,14]. Accordingly, we performed exercise training on high-fat *Drosophila* to explore how exercise training might impact high-fat diet-induced cardiac dysfunction.

As an important target in regulating systemic lipid metabolism, *mtp*, when expressed in the liver and adipose, is involved in the correct folding and assembly of downstream apoB, affecting the secretion of chylomicrons (CMs) and the synthesis of very low-density lipoproteins (VLDL) [3]. In addition, *mtp* in cardiac myocytes is involved in regulating systemic lipid metabolism, and knocking down *mtp* in the heart alone can prevent the increase in systemic TG levels caused by HFD. Furthermore, knocking down *mtp* in the heart can protect systolic cardiac function from HFD to a certain extent, similar to treating *Drosophila* with HFD by exercise training. Although it is well-recognized that the heart plays an important role in regulating systemic lipid metabolism, and previous studies have found that *mtp* in cardiac myocytes can regulate systemic lipid metabolism under HFD conditions, the exact mechanism remains unclear [41].

Current evidence suggests that HFD induces myocardial hypertrophy and fibrosis, reduced coronary reserve and cardiac function. Importantly, exercise limits lipid metabolism disorders, cardiac hypertrophy and fibrosis and helps prevent lipotoxic cardiomyopathy. It is well-established that exercise lowers the heart rate [42] and alleviates cardiac oxidative stress by inhibiting LOX-1 receptor expression [43]. In addition, exercise-based cardiac rehabilitation reduces hospitalization rates and cardiovascular mortality and improves the quality of life [44]. A study found that exercise-based cardiac rehabilitation reduced hospitalization rates, cardiovascular mortality and improved quality of life. Interestingly, exercise was found to mediate cardioprotection through upregulation of miR-344g-5p, which targets Hmgcs2 mRNA, and inhibits HMGCS2 upregulation, thereby suppressing lipotoxicity [45]. It has also been suggested that aerobic exercise reverses cardiac remodeling by reducing inflammation, fibrosis and apoptosis in HFD rats, to a certain extent by inhibiting P2X7R expression in cardiomyocytes [46]. Herein, we showed that exercise training had a therapeutic effect on the decrease in cardiac *mtp* expression caused by HFD, both in *w^1118^*, *Hand > w^1118^* and *Hand > mtp^RNAi^ Drosophila*; Had1 and Acox3mRNA expression in the heart of HFD flies was elevated to some extent after exercise training, suggesting that the ability of cardiomyocytes to catalyze long-chain fatty acid catabolism may have increased.

## 5. Conclusions

Overall, we provided compelling evidence that knocking down *mtp* in the heart could prevent an increase in systemic TG levels and protect cardiac contractility from HFD-induced damage, which is similar to performing exercise training to some extent. Moreover, exercise training upregulated the decrease in cardiac *mtp* expression induced by HFD. An increase in the expression of Had1 and Acox3, which are downstream of *mtp,* was observed, which was highly consistent with the changes in cardiac *mtp*. This finding suggests that the upregulation of cardiac *mtp* by exercise training may be an important pathway to treat HFD-induced cardiac dysfunction and abnormal lipid metabolism. It is conceivable that elevated cardiac *mtp* expression indirectly drives up downstream β oxidase, thereby perhaps increasing the ability of cardiomyocyte mitochondria to break down triglycerides. Given the important role of cardiomyocyte *mtp* in regulating systemic lipid metabolism and protecting the heart from lipotoxicity, the ameliorative effects of exercise training on cardiac *mtp* provide new insights for future clinical studies related to lipotoxic cardiomyopathy, especially on the potential use of exercise training to treat abnormalities in lipid metabolism and cardiac dysfunction due to HFD.

## Figures and Tables

**Figure 1 biology-11-01745-f001:**
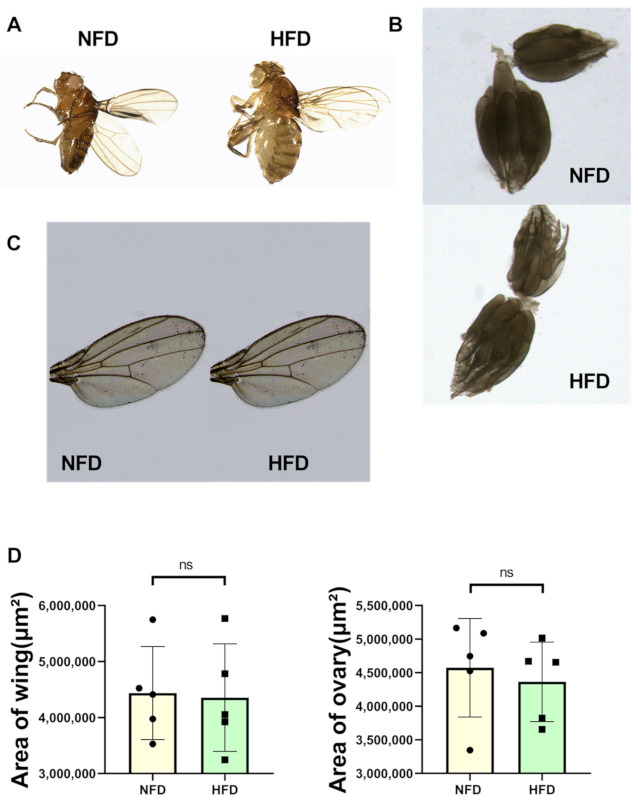
The *w^1118^* group and Hand > *w^1118^* group exhibited obesity after 5d of HFD feeding. (**A**) Morphological pictures of 10-day-old *Drosophila* under NFD and HFD feeding conditions, with HFD *Drosophila* being larger and having a protruding abdomen. (**B**) Photographs of ovaries of 10-day-old NFD and HFD flies. (**C**) Photographs of wings of 10-day-old NFD and HFD flies. (**D**) Comparison of wing size and ovary size between 10-day-old NFD and HFD flies (N = 5) using independent sample *t*-test.

**Figure 2 biology-11-01745-f002:**
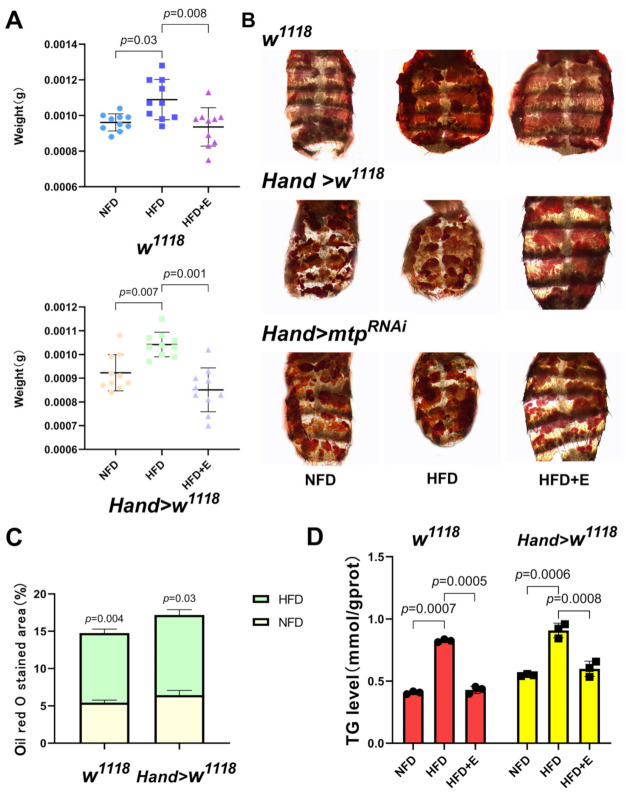
(**A**) Body weights were obtained by weighing the 10-day-old flies on an electronic microbalance, N = 10, with a significant increase in body weight in *Drosophila* after HFD feeding compared to the NFD group but a decrease in body weight after the exercise intervention. (**B**) ORO abdominal staining of each group of the 10-day-old flies under different treatment conditions. (**C**) ORO staining in the abdomen of *Drosophila*. N = 5. The area of ORO staining in the abdomen of *Drosophila* was larger in the HFD group compared to the NFD group. (**D**) Whole-body TG levels in 10-day-old *Drosophila*. Whole-body TG levels were significantly higher in the HFD group than in the NFD group. N = 10, repeated three times. One-way ANOVA was used for (**A**,**D**). An independent sample *t*-test was used for (**C**).

**Figure 3 biology-11-01745-f003:**
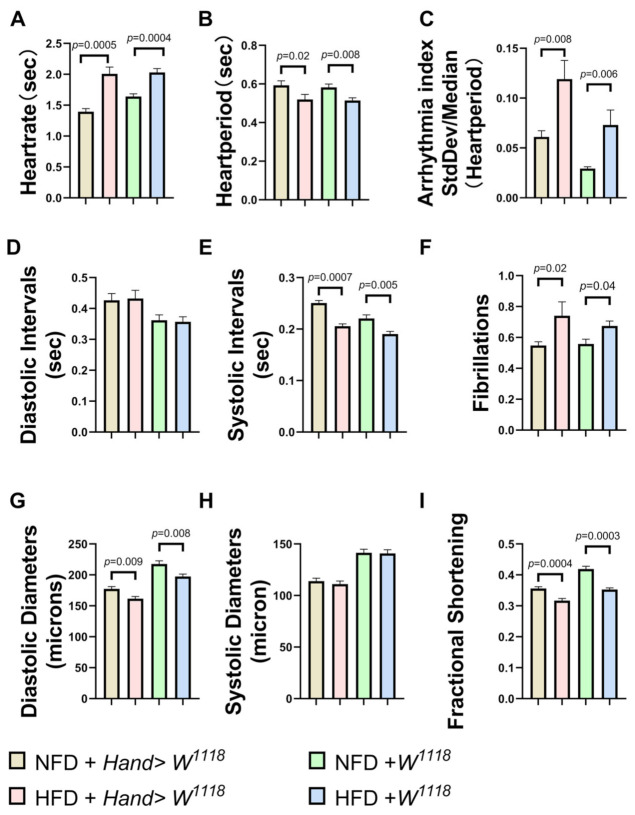
NFD and HFD groups of 10-day-old *Drosophila* M-mode ECG, with quantitative analysis of HR, HP, AI, DI, SI, FL, SD, DD, FS. Compared to NFD, *w^1118^* and Hand > *w^1118^* under HFD conditions, HR, AI and FL (**A**,**C**,**F**) were significantly upregulated. Moreover, HP, SI, DD and FS (**B**,**E**,**G**,**I**) decreased significantly, while DI and SD (**D**,**H**) did not show significant differences. All *Drosophila* were virgin flies, and the sample sizes for the two groups of NFD and HFD were N = 28 and N = 30, respectively. All *p* values were from independent samples *t*-tests.

**Figure 4 biology-11-01745-f004:**
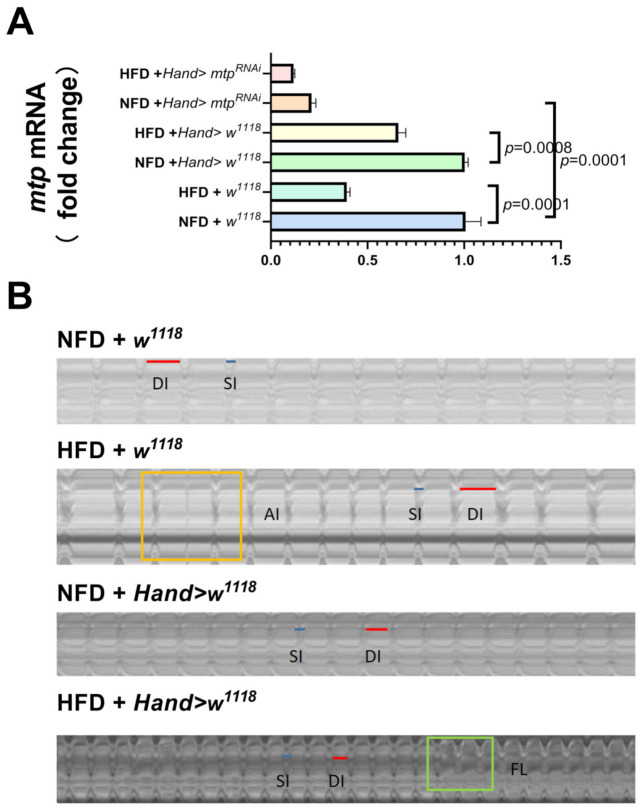
(**A**) Relative expression levels of *mtp* in *Drosophila* cardiomyocytes. *mtp* expression in *Drosophila* cardiomyocytes was significantly higher in the NFD group than in the HFD group. The test sample consisted of about 50 isolated hearts from the 10-day-old virgin flies. (**B**) M-mode ECGs under different treatments, the blue line is the systolic interval, and the red line is the diastolic interval; arrhythmia is marked in a yellow rectangle and fibrillation in a green rectangle. The intercept time for each group of M-mode ECGs was 10 s. In (A), one-way ANOVA was used to identify differences between the strains of *Drosophila*. LSD was used for post hoc testing.

**Figure 5 biology-11-01745-f005:**
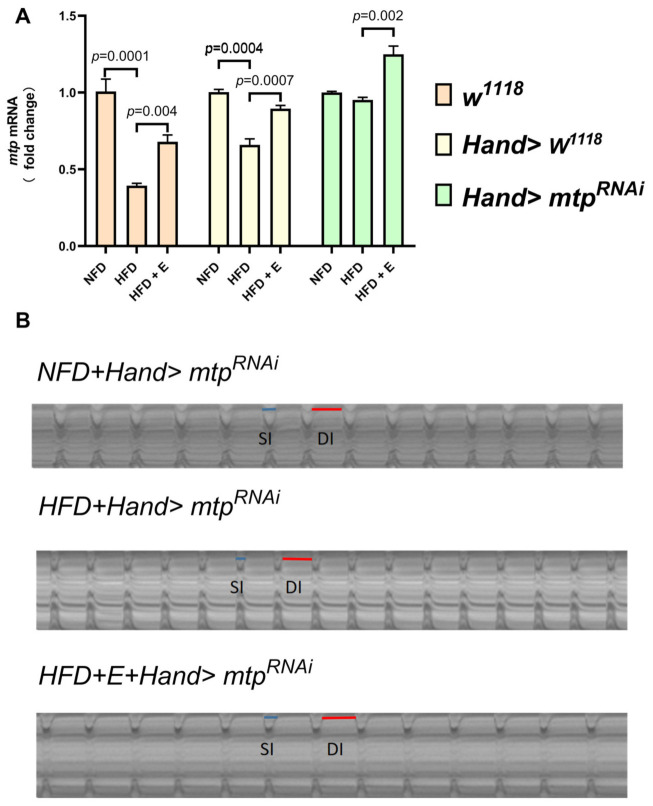
Effects of inhibition of cardiac *mtp* and exercise training on mRNA expression in *Drosophila*. All samples (N = 50 per group) were virgin flies of 10 days. (**A**) The cardiac mRNA expression levels of *Drosophila* under different treatments, on the one hand, HFD decreased the mRNA expression levels of cardiac *mtp* in the *w^1118^* and *Hand > w^1118^* groups; on the other hand, HFD did not significantly affect the mRNA expression levels of cardiac *mtp* in *Hand > mtp^RNAi^ Drosophila*. In contrast, exercise training in *Drosophila* with HFD upregulated the mRNA expression levels of cardiac *mtp* in all *Drosophila* of HFD. (**B**) M-mode ECG of *Drosophila* in the *Hand > mtp^RNAi^* group, the blue line is the systolic interval, and the red line is the diastolic interval. In (A), one-way ANOVA was used to identify differences between strains of *Drosophila* NFD, HFD and HFD +E. LSD was used for post hoc testing.

**Figure 6 biology-11-01745-f006:**
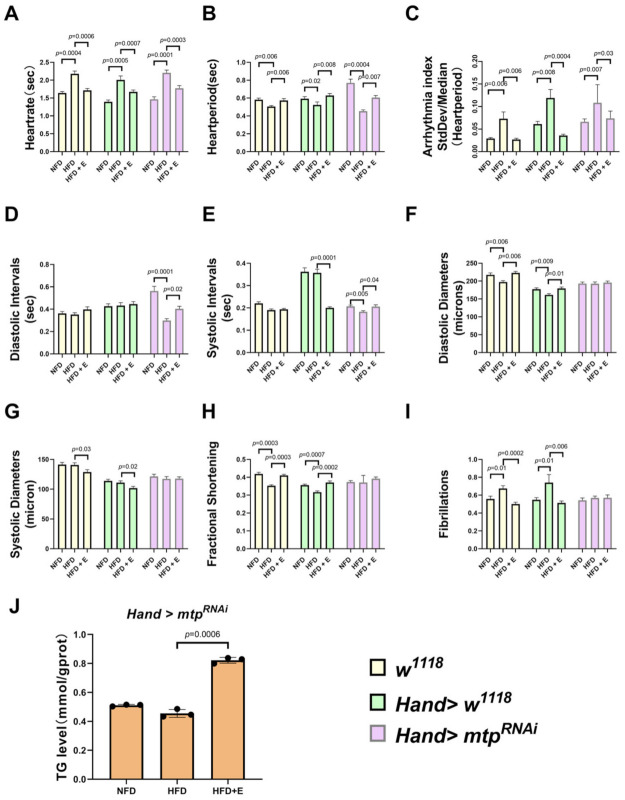
Effect of inhibition of cardiac *mtp* and exercise training on cardiac function and *Hand > mtp^RNAi^* whole-body TG levels in *Drosophila*. (**A**–**I**) Under different treatments, cardiac function assays included HR, HP, AI, DI, SI, DD, SD, FS and FL. NFD + *w^1118^*, HFD + *w^1118^*, HFD + E+*w^1118^*, NFD + Hand > *w^1118^*, HFD + Hand > *w^1118^*, HFD + E + Hand > *w^1118^*, NFD + *Hand > mtp^RNAi^*, HFD + *Hand > mtp^RNAi^*, and HFD + E + *Hand > mtp^RNAi^*. Sample sizes N = 26, 28, 24, 27, 28, 19, 34, 22 and 22. All samples were virgin flies of 10 days. (**J**) Whole-body TG levels of *Hand > mtp^RNAi^ Drosophila*. All samples were virgin flies of 10 days, N = 10, and measurements were repeated three times. Whole-body TG levels were significantly higher in *Drosophila* in the HFD + E group compared to the HFD group. One-way ANOVA was used to identify differences between the strains of *Drosophila* NFD, HFD and HFD + E. LSD was used for post hoc testing.

**Figure 7 biology-11-01745-f007:**
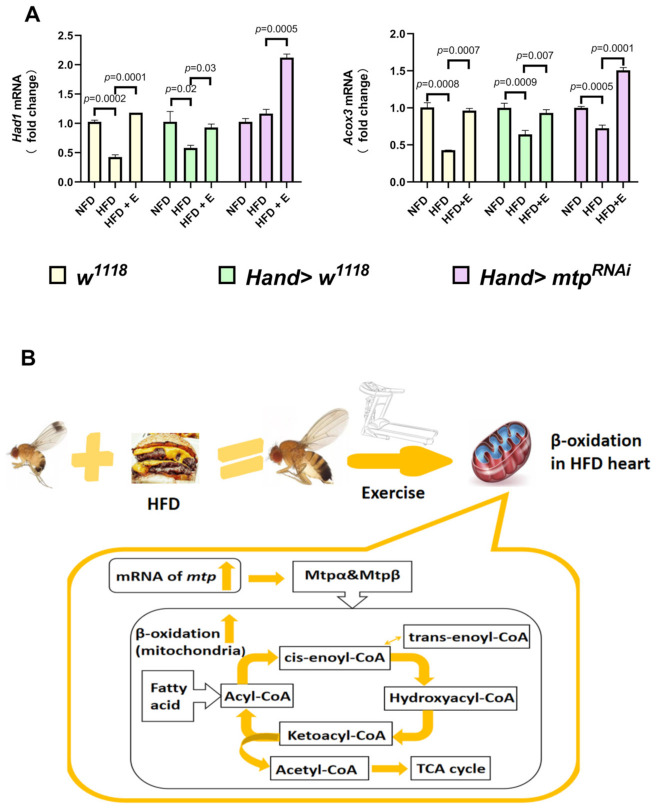
(**A**) Comparison of the expression levels of Had1 and Acox3mRNA in *w^1118^*, *Hand > w^1118,^* and *Hand > mtp^RNAi^* hearts under different treatments using one-way ANOVA with LSD for post hoc testing. Compared to the NFD groups, Had1 expression was decreased in HFD + *w^1118^* and HFD + *Hand > w^1118^ Drosophila* cardiomyocytes, while Acox3 expression was decreased in HFD + *w^1118^*, HFD + *Hand > w^1118^* and HFD + *Hand > mtp^RNAi^ Drosophila* cardiomyocytes. Compared with the HFD groups, there was a general increase in Had1 and Acox3 in *w^1118^*, *Hand > w^1118,^* and *Hand > mtp^RNAi^ Drosophila* cardiomyocytes after exercise training. All samples were virgin flies of 10 days (N = 50 per group). (**B**) Putative role of *mtp* in β-oxidation in the heart stimulated by exercise training.

## Data Availability

Data are contained within the article.

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
