# Peer review of "Exercise Training Upregulates Cardiac mtp Expression in Drosophila melanogaster with HFD to Improve Cardiac Dysfunction and Abnormal Lipid Metabolism"

_biology, 2022, doi:10.3390/biology11121745_

Round 1
Reviewer 1 Report
The heart regulates systemic lipid metabolism, and a high fat diet leads to increased systemic lipid accumulation and cardiac dysfunction. Peng and colleagues established a Drosophila melanogaster model of HFD feeding-induced cardiac dysfunction and abnormal lipid metabolism in this study. They investigated whether exercise training in a high-fat diet (HFD) Drosophila can improve systemic lipid overload and cardiac dysfunctions accompanied by altered expression of certain genes.
They claim that they found significant sex differences before and after exercise training, although they report results only in female flies.
They had knockdown Mtp gene and found a positive health effect in HFD-induced elevation of systemic triglycerides and improved heart contractility. Exercise training with starvation led to rescue or upregulating the expression of two genes, Had1 and Acox3, involved in the ß-oxidation of long-chain fatty acids and ameliorating the impaired systemic lipid metabolism and cardiac function by HFD.
The MTP lipid transfer activity assembles and contributes to the secretion of ApoB family lipoproteins that carry lipids between different tissues. How the heart influences mitochondrial β-oxidation by regulating the expression of Mtp to affect systemic lipid metabolism is not discussed nor clarified in this paper.
Major issues and comments:
- Given that it only reports female flies, the statement in the abstract must be removed" We found significant sex differences before and after exercise training."
- Are authors using virgin or single-mated females? This is only mentioned in one experiment, but it is nuclear in methods or if this is general for all the study.
- Reference Stobdan et al. 2019 should be included and discussed.
- To control for body weight due to HFD, please include wing size. The wing should stay the same. And the size of the ovary to examine whether the increased size ovary due to lipid uptake contributes to the bulging abdomen in the HFD.
- The authors apparently wrongly state, "mtp in drosophila cardiomyocytes was significantly lower in the NFD group than in the HFD group". The graph shows the opposite trend, with mtp expression being reduced in the HFD group.
- Figure 6J: Why does exercise increase TAG levels in Hand>mtp-RNAi?
- 0.8 level is a value as high as an HFD without exercise and mtp-RNAi. This observation may contradict the data that HDF+E decreased body weight in figure 1.
- Please correct typos and grammatical errors throughout the text. Drosophila should be a capital letter, and w1118 should be lowercase.
- "These results suggest increased dietary fat leads to obesity and cardiac dysfunction in Drosophila, and this mechanism may be related to reduced mtp in cardiac myocytes." The above contradicts the subtitle "Specific knockdown of mtp in the heart alleviates HFD-induced cardiac dysfunction, which is similar to the effect of exercise training". It also disagrees with the findings that exercise increases mtp expression levels. This statement needs to be clarified.
- Exercise training reduces whole-body TG levels in HFD+W1118 and HFD+Hand>W1118 but significantly increases in HFD with Hand>mtp-RNAi. Why? And how?
- Suppose HFD decreases mtp and increases systemic TAG. In that case, it is seemingly contradictory that knockdown of mtp in the heart precluded the increase in systemic TG levels and that exercise-mediated increase in mtp had the same effect as silencing.
- The KD of mtp and exercise lead to ameliorating HFD-induced cardiac dysfunction. Still, the mechanism may not be the same as suggested, given the results in Figure 6J and the effects of exercise on Mtp gene expression.
- Figure 5A is truncated in the manuscript I have for review. Therefore I cannot judge this statement "HFD+Hand>mtpRNAi group of drosophila showed a significant increase in cardiac mtp expression. Yet, how could a condition with mtp-RNAi have increased cardiac mtp?
- If mtp-RNAi rescues lipid overload, how can exercise rescue the decrease in Had1 expression and the decrease in Acox3 expression but increases mtp? If exercise training reverses the decrease in cardiac β-oxidation induced by HFD also in Hand>mtp-RNAi, do the two interventions act independently.
- The authors state, "knocking down mtp in the heart can protect systolic cardiac function from HFD to a certain extent. This effect is similar to the treatment of drosophila melanogaster with HFD by exercise training. However, some observations do not agree with this statement. Exercise increases mtp expression and massively increases TAG levels in Hand>mtp-RNAi condition.
- The authors also state, "The increased ability to catalyse long-chain fatty acid catabolism prevents excessive accumulation of lipids in the heart and protects the heart from damage." This statement is unfounded based on Figure 6J. Yet, it is interesting. Please comment and clarify Figure 6J.
In conclusion while there is some or extensive rescue of the cardiac function in exercised flies and Hand>mtp-RNAi, the exact mechanism relating exercise with HFD-regulated mtp levels still needs clarification. A lacking experiment that could clarify the mechanism is overexpressing mtp with Hand-Gal4 in the HFD group with or without exercise.
Author Response
Responds to the reviewer’s comments:
Q1. Given that it only reports female flies, the statement in the abstract must be removed" We found significant sex differences before and after exercise training."
A1. Thank you very much for pointing out the problem.We are extremely sorry that there is a sentence error in the sentence that" We found significant sex differences before and after exercise training."We have deleted it in the corrected essay .
Q2. Are authors using virgin or single-mated females? This is only mentioned in one experiment, but it is nuclear in methods or if this is general for all the study.
A2. Thank you very much for your question. All experimental samples were virgin flies in this research. This is general for the most similar studies.
Q3.Reference Stobdan et al. 2019 should be included and discussed.
A3.Thank you very much for the references you provided, which we have included in the paper(in reference 26).
Q4.To control for body weight due to HFD, please include wing size. The wing should stay the same. And the size of the ovary to examine whether the increased size ovary due to lipid uptake contributes to the bulging abdomen in the HFD.
A4.Thanks for your detailed advice .We have added these materials in the corrected essay (in figure1 B,C and D).
Q5.The authors apparently wrongly state, "mtp in drosophila cardiomyocytes was significantly lower in the NFD group than in the HFD group". The graph shows the opposite trend, with mtp expression being reduced in the HFD group.
A5.We are so sorry for this written error which should not happened in our paper. We have corrected it in the corrected essay .
Q6.Figure 6J: Why does exercise increase TAG levels in Hand>mtp-RNAi?0.8 level is a value as high as an HFD without exercise and mtp-RNAi. This observation may contradict the data that HDF+E decreased body weight in figure 1.
A6.Thank you very much for your question. In our viewpoint ,mtp plays an important role in lipid synthesis. After exercise training, the increase of mtp mRNA expression level in Hand>mtp-RNAi was significantly greater than that of other mtp non-knockdown groups,so the level of lipid synthesis in Hand>mtp-RNAi flies was greatly enhanced, and the corresponding TG level of flies in the mtp-RNAi group was higher than that in the control group. Thus this observation does not contradict with the data that HDF+E decreased body weight in figure 1,which deta are from normal flies.
Q7. Please correct typos and grammatical errors throughout the text. Drosophila should be a capital letter, and w1118 should be lowercase.
A7. We pretty apologize for the grammar and typing errors in the article, which we have corrected in time.Drosophila have been a capital letter, and w1118 have been lowercase.
Q8. "These results suggest increased dietary fat leads to obesity and cardiac dysfunction in Drosophila, and this mechanism may be related to reduced mtp in cardiac myocytes." The above contradicts the subtitle "Specific knockdown of mtp in the heart alleviates HFD-induced cardiac dysfunction, which is similar to the effect of exercise training". It also disagrees with the findings that exercise increases mtp expression levels. This statement needs to be clarified.
A8. Thank you very much for your question. Mtp is an essential factor in the assembly and secretion of the major apoB-lipoproteins termed lipophorins or Lpp in flies .Normally increased dietary fat leads to obesity and cardiac dysfunction in Drosophila, and this mechanism may be related to reduced mtp in cardiac myocytes,but after the specific knockdown of mtp in the heart,the regulation of systemic lipid metabolism is largely affected,especially on high-fat diet,leading to impaired triglyceride synthesis and preventing excess lipid accumulation in the heart. This result is similar to exercise training heightening systemic lipid metabolism,which accelerates the metabolism of triglycerides so that triglycerides do not build up excessively in the heart and affect cardiac function.
Q9. Exercise training reduces whole-body TG levels in HFD+W1118 and HFD+Hand>W1118 but significantly increases in HFD with Hand>mtp-RNAi. Why? And how?
A9.Thank you very much for your question. In our viewpoint, mtp plays an important role in lipid synthesis. After exercise training, the increase of mtp mRNA expression level in Hand>mtp-RNAi was significantly greater than that of other mtp non-knockdown groups,so the level of lipid synthesis in Hand>mtp-RNAi flies was greatly enhanced, and the corresponding TG level of flies in the mtp-RNAi group was higher than that in the control group.
Q10.Suppose HFD decreases mtp and increases systemic TAG. In that case, it is seemingly contradictory that knockdown of mtp in the heart precluded the increase in systemic TG levels and that exercise-mediated increase in mtp had the same effect as silencing.
A10.Thank you very much for your question. Normally increased dietary fat leads to obesity and cardiac dysfunction in Drosophila, and this mechanism may be related to reduced mtp in cardiac myocytes,but after the specific knockdown of mtp in the heart,the regulation of systemic lipid metabolism is largely affected,especially on high-fat diet,leading to impaired triglyceride synthesis and preventing excess lipid accumulation in the heart. Exercise mediated increases in mtp and mtpKD only have a similar effect in preventing the damage of HFD, and exercise training is significantly more effective in the treatment of HFD than mtpKD. Knocking down cardiac mtp could reduce the damage of myocardial contractile function (DD, SD, FL and FS) by HFD to some extent, but after exercise training, HFD+E+W1118 and HFD+E+Hand>W1118 flies showed slower heart rate, increased cardiac period, decreased arrhythmia index, increased diastolic diameter, decreased systolic diameter, reduced fibrillation and increased shortening fraction (Figure 6A, B, C, F, G, H and I), with significant improvements in both cardiac rhythm and systolic function, effectively treating HFD-induced cardiac dysfunction.
Q11. The KD of mtp and exercise lead to ameliorating HFD-induced cardiac dysfunction. Still, the mechanism may not be the same as suggested, given the results in Figure 6J and the effects of exercise on Mtp gene expression.
A11.Thank you very much for your question. Admittedly The KD of mtp and exercise lead to ameliorating HFD-induced cardiac dysfunction in normal groups . According to Figure 6J ,however,mtp gene expression in Hand>mtp-RNAi is abnormally expressed ,which is relatively lower than others. So the effects of exercise on mtp gene expression will lead to different influence.
Q12. Figure 5A is truncated in the manuscript I have for review. Therefore I cannot judge this statement "HFD+Hand>mtpRNAi group of drosophila showed a significant increase in cardiac mtp expression. Yet, how could a condition with mtp-RNAi have increased cardiac mtp?
A12.We are pretty apologize for it . Perhaps there are some mistakes in the manuscript ,and
we have corrected them in the corrected essay . Furthermore exercise training can increase mtp expression levels of HFD+Hand>mtpRNAi group.
Q13. If mtp-RNAi rescues lipid overload, how can exercise rescue the decrease in Had1 expression and the decrease in Acox3 expression but increases mtp? If exercise training reverses the decrease in cardiac β-oxidation induced by HFD also in Hand>mtp-RNAi, do the two interventions act independently.
A13. Exercise training increases the expression of mtp, and the up-regulation of mtp promotes the expression of downstream Acox3. The present study failed to elucidate how exercise rescued decreased Had1 expression, but only speculated about an increase in β-oxidative capacity through increased Had1 and Acox3 expression.The changes in mtp expression are consistent with the changes in downstream Acox3 expression, so we suggest that exercise training upregulates mtp expression first, and the increase in mtp expression then promotes the increase in downstream Acox3 expression. There is no direct relationship between Had1 and mtp, but the β-oxidation capacity can be evaluated by the expression of Had1.
Q14. The authors state, "knocking down mtp in the heart can protect systolic cardiac function from HFD to a certain extent. This effect is similar to the treatment of drosophila melanogaster with HFD by exercise training. However, some observations do not agree with this statement. Exercise increases mtp expression and massively increases TAG levels in Hand>mtp-RNAi condition.
A14. Thank you very much for your question. "Knocking down mtp in the heart can protect systolic cardiac function from HFD to a certain extent"is a protected pathway by blocking lipid synthesis .Exercise training reduces whole-body TG levels in HFD+W1118 and HFD+Hand>W1118 to protect cardiac function ,which is another protected pathway by speeding up lipid metabolism. And mtp plays an important role in lipid synthesis. After exercise training, the increase of mtp mRNA expression level in Hand>mtp-RNAi was significantly greater than that of other mtp non-knockdown groups,so the level of lipid synthesis in Hand>mtp-RNAi flies was greatly enhanced, and the corresponding TG level of flies in the mtp-RNAi group was higher than that in the control group.
Q15. The authors also state, "The increased ability to catalyse long-chain fatty acid catabolism prevents excessive accumulation of lipids in the heart and protects the heart from damage." This statement is unfounded based on Figure 6J. Yet, it is interesting. Please comment and clarify Figure 6J.
A15.Thank you very much for pointing out the problem. This statement indeedly is unfounded based on Figure 6J,which is our error of expression and have been corrected in revised paper. In this study, β-oxidation function was assessed by β-hydroxy acid dehydrogenase 1 (Had1) and 3-hydroxy Acyl-CoA oxidase 3 (Acox3), drawing on previous methods.Exercise training increased the expression levels of Had1 and Acox3,and the increased expression of Had1 and Acox3 indicates an increased capacity for β-oxidation, which primarily breaks down long-chain fatty acids,.Thus exercise training improves the ability to catalyze the breakdown of long-chain fatty acids, prevents excessive accumulation of lipids in the heart, and protects the heart from damage.

Reviewer 2 Report
This is a generally well-performed study looking at exercise effects upon Drosophila fed a high-fat diet, with a focus on heart function and proteins involved in fat metabolism. The conclusions are generally sound, but there is some concern about the role of mtp expression levels, given that knockdown with RNAi or increase with exercise seem to have similar positive effects. The authors need to make address this point more clearly (as it written in a confusing manner) and offer some explanation as to molecular mechanisms involved. Below are suggestions for improvement.
Major Concerns
1. Figure 3, 4, 5 legends. Why is this indicated as an “M-mode ECG”? Is it not a kymograph of single pixel images?
2. Why were only female flies analyzed? If this is the case, why does the Simple Summary state: “We found significant sex differences before and after exercise training.”
3. How do the authors know that the mtp expression is within cardiomyocytes, as claimed several times? While the mtp knockdown is cardiomyocyte specific, the RNA expression is presumably measured in whole hearts, which include pericardial cells and the skeletal muscle ventral layer.
4. The paragraph that discusses mtpRNAi effects is confusing. The authors need to more clearly explain the effects of RNAi and how it correlates with the observed phenotypes. Further, they need to more clearly explain why mtp knockdown causes a positive effect while downregulation of mtp caused by HFD yields a negative effect. Likewise, this statement in the conclusions is confusing: “Exercise training upregulated the decrease in cardiac mtp expression induced by HFD”
5. Were effects on longevity or heart myofibrillar structure studied?
Minor Concerns
1. Drosophila should be capitalized and italicized. melanogaster should be italicized but not capitalized.
2. Define mtp in the abstract.
3. Remove “induced” from line 25.
4. Line 46, “(ei.” should be “(i.e.,”
5. Why are celiac particles abbreviated as CM?
6. Lines 68-70: Duplicated text: “It also inhibited inflammation in the adipose tissue of obese mice induced by a high-fat diet.[10][11] It also inhibits inflammation in the adipose tissue of obese mice on a high-fat diet.” Also, lines 84-85 contains duplicated text: “However, the specific mechanisms need to be further explored. However, the exact mechanism needs to be further explored.”
7. Line 74, delete “for fruit flies”
8. Regarding heart function tests (line 197 etc.): fly hearts were not dissected from the organism. They were exposed via dissection. Change “(SOHA).” to “(SOHA) to assess”. The final sentence of the paragraph is redundant. What are “flight heartbeats”?
9. Oil Red O Dye (line 137, etc.). Confusingly written as passive and active tenses are combined in a single paragraph.
10. Line 184, change “cardiac cycle (HP), arrhythmia (AI), diastolic (DI) and systolic (SI)” to “cardiac cycle or heart period (HP), arrhythmia (AI), diastolic interval (DI) and systolic interval (SI)”
11. Please state the ages of the flies analyzed in the figure legends.
12. The right-most panel in Figure 5 is cut off. In this regard, the following statement in the legend should indicate in what direction mRNA expression is affected: “on the other hand, HFD significantly affected the mRNA expression levels of cardiac mtp in Hand>mtpRNAi drosophila” This legend also indicates panel “C”, where it should be “B”.
13. Figure 3 needs to be mentioned in the text (page 6).
14. Why does reference to Figure 6 occur prior to reference to Figure 5?
15. Why are sample sizes given in some legends, but not others?
16. This is an overstatement (line 313): “drosophila has a heart structure similar to that of vertebrates”
Author Response
Q1.Figure 3, 4, 5 legends. Why is this indicated as an “M-mode ECG”? Is it not a kymograph of single pixel images?
A1.Thank you very much for your question. We provide a detailed reference for your question .(Fink M, Callol-Massot C, Chu A, Ruiz-Lozano P, Izpisua Belmonte JC, Giles W, Bodmer R, Ocorr K. A new method for detection and quantification of heartbeat parameters in Drosophila, zebrafish, and embryonic mouse hearts. Biotechniques. 2009 Feb;46(2):101-13. doi: 10.2144/000113078. PMID: 19317655; PMCID: PMC2855226.)
Q2. Why were only female flies analyzed? If this is the case, why does the Simple Summary state: “We found significant sex differences before and after exercise training.”
A2. Female flies have more advantages than male flies.We are extremely sorry that there is a word error in the sentence that We found significant sex differences before and after exercise training.The word sex is a written error.We have corrected it in the corrected essay .
Q3. How do the authors know that the mtp expression is within cardiomyocytes, as claimed several times? While the mtp knockdown is cardiomyocyte specific, the RNA expression is presumably measured in whole hearts, which include pericardial cells and the skeletal muscle ventral layer.
A3. The UAS/Gal4 system is commonly used in Drosophila to manipulate a specific cell type or tissue to determine the requirements for genes and pathways either in regulating that same cell type/tissue of interest or in remotely affecting separate tissues (Brand and Perrimon 1993). The UAS/Gal4 system employs the yeast transcription factor Gal4 under the control of a “tissue-specific” enhancer/promoter sequence (referred to as the “driver”) in combination with a “responder” that contains an Upstream Activating Sequence composed of Gal4 binding sites upstream of a target gene or sequence of interest (Brand and Perrimon 1993). Gal4 binds to the UAS sequence, thereby inducing tissue-specific expression of the transgene (e.g., fluorescent reporter, hairpin RNA, protein-coding gene, etc). There are some detailed references.(1.Weaver LN, Ma T, Drummond-Barbosa D. Analysis of Gal4 Expression Patterns in Adult Drosophila Females. G3 (Bethesda). 2020 Nov 5;10(11):4147-4158. doi: 10.1534/g3.120.401676. PMID: 32917721; PMCID: PMC7642949. 2.Brand AH, Perrimon N. Targeted gene expression as a means of altering cell fates and generating dominant phenotypes. Development. 1993 Jun;118(2):401-15. doi: 10.1242/dev.118.2.401. PMID: 8223268.)
Q4. The paragraph that discusses mtpRNAi effects is confusing. The authors need to more clearly explain the effects of RNAi and how it correlates with the observed phenotypes. Further, they need to more clearly explain why mtp knockdown causes a positive effect while downregulation of mtp caused by HFD yields a negative effect. Likewise, this statement in the conclusions is confusing: “Exercise training upregulated the decrease in cardiac mtp expression induced by HFD”
A4. (1)RNA interference (RNAi) refers to a gene silencing phenomenon induced by double-stranded RNA in molecular biology, and its mechanism is to inhibit gene expression by hindering the transcription or translation of specific genes.The GAL4-UAS system is a biochemical method used to study gene expression and function in organisms such as Drosophila melanogaster. Gal4 binding to the UAS sequence activated gene expression. This method was introduced in Drosophila in 1993 by Andrea Brand and Norbert Perrimon and is considered a powerful technique for studying gene expression. The system consists of two parts: the Gal4 gene, encoding the yeast transcriptional activation protein Gal4, and the UAS (upstream activation sequence), enhancers that GAL4 binds specifically to activate gene transcription. (2)The knockout of mtp in the heart by this technique resulted in reduced mtp expression in the heart of Drosophila, and the flies could survive, but with shortened lifespan, decreased motility and reduced number of eggs laid.(3)For mtpKD flies, their ability to synthesize lipids is much lower than that of normal flies, so their damage to HFD is reduced. However, HFD can down-regulate the expression of mtp in the heart of normal flies, and further affect their lipid metabolism and cardiac function. (4)After exercise intervention, the expression of mtp in the heart of HFD flies is up-regulated, and the ability of systemic lipid metabolism is enhanced, indicating that exercise training can reverse the HFD-induced down-regulation of cardiac mtp expression.
Q5. Were effects on longevity or heart myofibrillar structure studied?
A5. We have not yet researched these effects, but we are also interested in these effects and would like to study these effect in next essay.
Minor Concerns
Q1. Drosophila should be capitalized and italicized. melanogaster should be italicized but not capitalized.
A1. Thanks for your kind reminders.We have corrected it in the corrected essay .
Q2. Define mtp in the abstract.
A2. Thanks for your kind advice.We have added it in the corrected essay .
Q3. Remove “induced” from line 25.
A3. Thanks for your kind advice.We have corrected it in the corrected essay .
Q4. Line 46, “(ei.” should be “(i.e.,”
A4. Thanks for your kind advice.We have corrected it in the corrected essay .
Q5. Why are celiac particles abbreviated as CM?
A5.We are so sorry for this mistake .Celiac particles should be replaced by chylomicrons (CMs).
Q6. Lines 68-70: Duplicated text: “It also inhibited inflammation in the adipose tissue of obese mice induced by a high-fat diet.[10][11] It also inhibits inflammation in the adipose tissue of obese mice on a high-fat diet.” Also, lines 84-85 contains duplicated text: “However, the specific mechanisms need to be further explored. However, the exact mechanism needs to be further explored.”
A6. We are so sorry for these mistakes and have corrected it in the corrected essay .
Q7.Line 74, delete “for fruit flies”
A7. Thanks for your kind advice.We have corrected it in the corrected essay .
Q8. Regarding heart function tests (line 197 etc.): fly hearts were not dissected from the organism. They were exposed via dissection. Change “(SOHA).” to “(SOHA) to assess”. The final sentence of the paragraph is redundant. What are “flight heartbeats”?
A8. Thanks for your kind advice.We have corrected it in the corrected essay , and“flight heartbeats”should be heartbeats.
Q9. Oil Red O Dye (line 137, etc.). Confusingly written as passive and active tenses are combined in a single paragraph.
A9. We are very sorry for the inconvenience caused by these errors,and have corrected them in the corrected essay.
Q10. Line 184, change “cardiac cycle (HP), arrhythmia (AI), diastolic (DI) and systolic (SI)” to “cardiac cycle or heart period (HP), arrhythmia (AI), diastolic interval (DI) and systolic interval (SI)”
A10.Thanks for your kind advice.We have corrected it in the corrected essay .
Q11. Please state the ages of the flies analyzed in the figure legends.
A11. This was an oversight on our part, and we have supplemented this information in the revised paper.
Q12. The right-most panel in Figure 5 is cut off. In this regard, the following statement in the legend should indicate in what direction mRNA expression is affected: “on the other hand, HFD significantly affected the mRNA expression levels of cardiac mtp in Hand>mtpRNAi drosophila” This legend also indicates panel “C”, where it should be “B”.
A12. We are so sorry for this mistake. This was an oversight on our part, and we have corrected it in the revised paper.
Q13. Figure 3 needs to be mentioned in the text (page 6).
A13.Thank you very much for your advice, and we have corrected it in the corrected essay.
Q14. Why does reference to Figure 6 occur prior to reference to Figure 5?
A14. We are so sorry for this mistake .We have corrected it in the corrected essay.
Q15.Why are sample sizes given in some legends, but not others?
A15.This was an oversight on our part.We have added other sample sizes in the corrected essay.
Q16. This is an overstatement (line 313): “drosophila has a heart structure similar to that of vertebrates”
A16.We are so sorry for this expressing error .We have corrected it in the corrected essay.

Reviewer 3 Report
Dear,
Manuscript Number: biology- 2020854
Title Manuscript: Exercise training upregulates cardiac mtp expression in drosophila melanogaster with HFD to improve cardiac dysfunction and abnormal lipid metabolism
This experimental study is an interesting and important issue since the authors investigated the molecular mechanisms of the heart in a species of fly called Drosophila melanogaster but at the moment MAJOR REVISIONS are necessary in order to make it suitable for a final decision for “Biology”.
POINTs of STRENGTH:
1) Assessment of the effect of exercise training intervention on molecular mechanisms of the heart in Drosophila melanogaster models (an experimental study);
2) Classified results in the results section;
POINTs of WEAKNESS (and/or should be revised to improve the manuscript):
Abstract:
3) The purpose of this experimental study is not specified in the objective section; please specify clearly;
4) The age, weight, gender of drosophila melanogaster, total groups and number of each group are not specified in the methods section; please clarify;
5) The study protocol and exercise training intervention (type and duration or follow-up) is not specified in the methods section; please clarify;
6) The significance level of the results section is not specified; please provide;
7) Please report the “conclusion section of the Abstract” based on the results obtained from the study.
Introduction:
8) Please provide the morphology of drosophila (Drosophila melanogaster); and provide the advantage of Drosophila melanogaster compared to other research models such as Mice and Zebrafish;
9) In addition, report the background literature on the effects of exercise training interventions on the molecular mechanisms of the heart or other tissues in the Drosophila species.
10) The hypothesis and purpose of this study can be stated in more detail;
2. Materials and methods
2.1. drosophila strains and husbandry
11) please clarify total fruit flies, number of groups, age, weight, gender, humidity of the storage place, circadian rhythm, and so on for the Drosophila species;
2.2. Heart function tests
12) This sentence has been repeated “arrhythmias, myocardial contractility, shortening fraction, etc.[26][27] The heart rate, cardiac cycle, arrhythmia, myocardial contractility, shortening fraction, etc. are analysed using semi-automatic optical heartbeat analysis software (SOHA).” Please remove;
2.4. Quantitative real-time fluorescence PCR (qPCR)
13) Please provide the Real-time PCR protocol, the temperature cycle of the genes and the melting curve shapes of genes and also the reference gene; In addition, please clarify the samples placed in Real-time PCR: Mono-pellet?, duplicate? Or triplicate?
2.7 Statistical analyses
14) Did authors use a statistical software to calculate the sample size? If YES, please explain and add its name and valid reference in the statistical analysis section.
15) Please clarify the significance level of the results for two-tailed? or one-tailed?
3. Results
16) The significance level of the results section is not specified; please clarify in the results section;
4. Discussion
17) Please add a valid reference for this sentence “Our further studies showed that exercise training had a therapeutic effect on the decrease in cardiac mtp expression caused by HFD, both in W1118 , Hand>W1118 and Hand>mtpRNAi drosophila, and that the expression of Had1 and Acox3mRNA in the heart of HFD drosophila was elevated to some extent after exercise training, suggesting that mitochondrial β-oxidation in cardiomyocytes.”
5. Conclusions
18) What are the conclusions and implications for future studies?;
19) What does this study add to the literature? Please clarify;
References
20) References section is not always in accordance with the authors' guidelines. In particular, please remove all brackets+numbers “[number]” in the references, and check No. 3 for validation.
Best Regards
11 November 2022
Author Response
Q3. The purpose of this experimental study is not specified in the objective section; please specify clearly;
A3.Thank you very much for your advice.We have added it in Line 29-31.
Q4. The age, weight, gender of drosophila melanogaster, total groups and number of each group are not specified in the methods section; please clarify;
A4. Thank you very much for your advice.We have added them in 2.1. Drosophila strains and husbandry.The weight is in figure2 A.
Q5. The study protocol and exercise training intervention (type and duration or follow-up) is not specified in the methods section; please clarify;
A5.Thank you very much for your question.We have corrected it in 2. Materials and Methods.
Q6.The significance level of the results section is not specified; please provide;
A6.We are so sorry for this mistake. This was an oversight on our part, and we have corrected it in the revised paper.
Q7. Please report the “conclusion section of the Abstract” based on the results obtained from the study.
A7.Thank you very much for your question.We have corrected it.
Q8.Please provide the morphology of drosophila (Drosophila melanogaster); and provide the advantage of Drosophila melanogaster compared to other research models such as Mice and Zebrafish;
A8.Thank you very much for your advice,and we have provided them in figure1A and line 74-84.
Q9.In addition, report the background literature on the effects of exercise training interventions on the molecular mechanisms of the heart or other tissues in the Drosophila species.
A9.References 31-33 are about the background literature on the effects of exercise training interventions on the molecular mechanisms of the heart or other tissues in the Drosophila species.
Q10.The hypothesis and purpose of this study can be stated in more detail;
A10.Thank you very much for your advice.We have corrected it in line 29-31.
Q11.please clarify total fruit flies, number of groups, age, weight, gender, humidity of the storage place, circadian rhythm, and so on for the Drosophila species;
A11.Thank you very much for your advice. We have added them in 2.1. Drosophila strains and husbandry, and the weight is in figure2 A.
Q12.This sentence has been repeated “arrhythmias, myocardial contractility, shortening fraction, etc.[26][27] The heart rate, cardiac cycle, arrhythmia, myocardial contractility, shortening fraction, etc. are analysed using semi-automatic optical heartbeat analysis software (SOHA).” Please remove;
A12. We are so sorry for this mistake. This was an oversight on our part, and we have corrected it in the revised paper.
Q13.Please provide the Real-time PCR protocol, the temperature cycle of the genes and the melting curve shapes of genes and also the reference gene; In addition, please clarify the samples placed in Real-time PCR: Mono-pellet?, duplicate? Or triplicate?
A13.Thank you very much for your advice.We have supplemented this information in 2.4. Quantitative real-time fluorescence PCR (qPCR).The samples placed in Real-time PCR are tested in triplicate.Melting curve shapes of genes are as follows:
Q14. Did authors use a statistical software to calculate the sample size? If YES, please explain and add its name and valid reference in the statistical analysis section.
A14.Yes,we added it in paragraph 2.7.
Q15.Please clarify the significance level of the results for two-tailed? or one-tailed?
A15.The significance level of the results is two-tailed, and we have corrected it in line 175.
Q16.The significance level of the results section is not specified; please clarify in the results section;
A16.We are so sorry for this mistake. This was an oversight on our part, and we have corrected it in the revised paper.
Q17.Please add a valid reference for this sentence “Our further studies showed that exercise training had a therapeutic effect on the decrease in cardiac mtp expression caused by HFD, both in W1118 , Hand>W1118 and Hand>mtpRNAi drosophila, and that the expression of Had1 and Acox3mRNA in the heart of HFD drosophila was elevated to some extent after exercise training, suggesting that mitochondrial β-oxidation in cardiomyocytes.”
A17.Thank you very much for your advice. We have corrected it in the revised paper.
Q18.What are the conclusions and implications for future studies?
A18.The improvement effect of exercise training on cardiac mtp may provide new insights into clinical research related to lipotoxic cardiomyopathy and potential future prospects for developing the use of exercise training to treat abnormal lipid metabolism and cardiac dysfunction due to HFD.
Q19.What does this study add to the literature? Please clarify;
A19.Given the important role of cardiomyocyte mtp in regulating systemic lipid metabolism and protecting the heart from lipotoxicity, the ameliorative effects of exercise training on cardiac mtp may provide new insights into clinical studies related to lipotoxic cardiomyopathy and potential future prospects for developing the use of exercise training to treat abnormalities in lipid metabolism and cardiac dysfunction due to HFD.
Q20.References section is not always in accordance with the authors' guidelines. In particular, please remove all brackets+numbers “[number]” in the references, and check No. 3 for validation.
A20.Thank you very much for your advice. We have corrected them in the revised paper.

Round 2
Reviewer 3 Report
Dear,
Manuscript Number: biology- 2020854
Title Manuscript: Exercise training upregulates cardiac mtp expression in Drosophila melanogaster with HFD to improve cardiac dysfunction and abnormal lipid metabolism
I am very grateful to the authors for their efforts.
In general, this manuscript has found suitable content after correcting major revisions, and the modified revisions are accepted.
Best Regards
22 November 2022